# Hospital-based caregiver intervention for people following hip fracture surgery (HIP HELPER): multicentre randomised controlled feasibility trial with embedded qualitative study in England

Toby O Smith [1,2] Reema Khoury,[3] Sarah Hanson [2] Allie Welsh [4] Kelly Grant,[3] Allan B Clark,[3] Polly-Anna Ashford,[3] Sally Hopewell [5] K Pfeiffer,[6] Phillipa Logan,[7] Maria Crotty,[8] Matthew L Costa,[5] Sarah Lamb,[5,9] The HIP HELPER Study Collaborators

For numbered affiliations see end of article.

**Correspondence to**
Professor Toby O Smith;
toby.smith@uea.ac.uk

## ABSTRACT

**Objectives** To assess the feasibility of conducting a pragmatic, multicentre randomised controlled trial (RCT) to test the clinical and cost-effectiveness of an informal caregiver training programme to support the recovery of people following hip fracture surgery.

**Design** Two-arm, multicentre, pragmatic, open, feasibility RCT with embedded qualitative study.

**Setting** National Health Service (NHS) providers in five English hospitals.

**Participants** Community-dwelling adults, aged 60 years and over, who undergo hip fracture surgery and their informal caregivers.

**Intervention** Usual care: usual NHS care. Experimental: usual NHS care *plus* a caregiver–patient dyad training programme (HIP HELPER). This programme comprised three, 1 hour, one-to-one training sessions for a patient and caregiver, delivered by a nurse, physiotherapist or occupational therapist in the hospital setting predischarge. After discharge, patients and caregivers were supported through three telephone coaching sessions.

**Randomisation and blinding** Central randomisation was computer generated (1:1), stratified by hospital and level of patient cognitive impairment. There was no blinding.

**Main outcome measures** Data collected at baseline and 4 months post randomisation included: screening logs, intervention logs, fidelity checklists, acceptability data and clinical outcomes. Interviews were conducted with a subset of participants and health professionals.

**Results** 102 participants were enrolled (51 patients; 51 caregivers). Thirty-nine per cent (515/1311) of patients screened were eligible. Eleven per cent (56/515) of eligible patients consented to be randomised. Forty-eight per cent (12/25) of the intervention group reached compliance to their allocated intervention. There was no evidence of treatment contamination. Qualitative data demonstrated the trial and HIP HELPER programme was acceptable.

**Conclusions** The HIP HELPER programme was acceptable to patient–caregiver dyads and health professionals. The

## STRENGTHS AND LIMITATIONS OF THIS STUDY

⇒ Mixed-method approach provided useful feasibility and acceptability data.
⇒ Assessment of diverse measures allowed evaluation of data collection for key outcome domains.
⇒ Participant experiences and acceptability data suggest perceived value in the HIP HELPER programme.
⇒ 10% of the cohort were living with cognitive impairment; none were recruited to the qualitative substudy.
⇒ COVID-19 pandemic affected National Health Service, which impacted on study delivery.

COVID-19 pandemic impacting on site's ability to deliver the research. Modifications are necessary to the design for a viable definitive RCT.

**Trial registration number** ISRCTN13270387.

## INTRODUCTION

Hip fracture is a serious injury for older people.[1] Approximately 80 000 people aged 60 years and over experience a fragility hip fracture in the UK annually.[2] This has an estimated combined health and social cost of over £2 billion.[3]

People have frequently experienced poor recovery following hip fracture.[4] The majority never return to preinjury levels of function.[3 5] Health-related quality of life (HRQoL) is reduced and mortality is high.[5 6] Patients also often experience repeated falls. This leads to reduced independence and confidence in self-caring skills. Approximately 20% of patients who previously lived at home move into institutional care following hip

fracture.[7] For those who do return home, informal caregivers who support their friend's/family member's care need frequently experience physical and mental stress.[4] A high caregiver burden that has previously been reported by 50%, 36% and 26% at 1 month, 3 months and 1 year post surgery[8] shows the multifaceted strain perceived by at least a subgroup of hip-fracture caregivers.

People after hip fracture who return home often need help. This ranges from assistance with personal activities of daily living (ADLs) such as toileting, washing, dressing and eating, to more complex tasks such as managing money, shopping and household chores.[9] Most of this required help is provided by family members or friends. Depending on the prefracture status of the patient, some of these informal caregivers continue in their caregiving role, others become a first-time caregiver.

While informal caregivers may be willing to support their friend/family member, they frequently feel underskilled, and have low confidence to do so.[10] A lack of information sharing, disorganised discharge planning and unclear individual roles have been identified as challenges for patients following hip fracture and their caregivers during care transitions.[11] Teaching caregiver skills to better support patients following hip fracture may improve HRQoL and independence, while reducing the burden of impairment for patients and caregivers.[10 12]

This study aimed to assess the feasibility of conducting a pragmatic, multicentre, randomised controlled trial (RCT) to test the clinical and cost-effectiveness of an informal caregiver training programme to support the recovery of people following hip fracture surgery.

## METHODS
The study was reported to satisfy the Consolidated Standards of Reporting Trials CONSORT) extension for reporting pilot and feasibility RCTs.[13] A full protocol has been published previously.[14] The study followed the published protocol with the exception of the introduction of the optional delivery of the HIP HELPER programme through an online approach rather than face-to-face delivery. This was in response to the COVID-19 pandemic, and enacted for one participant dyad.

### Study design
This was a feasibility study comprising a parallel, multicentre, pragmatic RCT and embedded qualitative study. The study process evaluation results are presented in this paper.

The study flow chart is presented as figure 1.

### Eligibility criteria
Participants were recruited from orthopaedic services in five National Health Service (NHS) hospitals in England providing hip fracture surgery. We recruited adults who previously lived in the community (not institutional care), aged 60 years and over, who had undergone hip fracture surgery, could nominate an informal caregiver and provided both patient–caregiver consent to participate. Where a patient–participant did not have capacity, agreement from a consultee was sought.

We excluded people who had acute, unstable or terminal illness or were expected by the clinical team to be discharged to a care home (residential or nursing). Caregivers were ineligible if they had an Abbreviated Mental Test Score (AMTS)[15] of less than 8.

### Study treatments
Usual NHS surgical and rehabilitation care was received by both control and intervention groups.[16] Accordingly, post-hip fracture surgery, all participants received predischarge care including nursing, physiotherapy, occupational therapy and social service needs-assessment (where appropriate). Patients and their caregivers in the control group did not receive the HIP HELPER programme, with no additional inpatient or outpatient caregiver training.

The HIP HELPER intervention has been previously described.[14] In brief, this was a patient–caregiver dyad training programme. The theoretical principle behind the programme is a social learning theory.[17]

In practice, people randomised to the experimental group received the usual NHS care in addition to the HIP HELPER programme. The only difference between the groups was the addition of three, 60 min, health professional–caregiver dyad HIP HELPER training sessions, performed in the hospital setting while the patient was an inpatient, and three follow-up telephone calls one, 3 and 6 weeks after hospital discharge. In the inpatient sessions, participants were taught about the normal recovery process, and skills in goal setting, pacing, activity behaviour modification and stress management. They were also taught skills on manual handling, transfers, walking and how to support people with ADLs. The follow-up telephone calls aimed to re-enforce the skills developed in the face-to-face sessions, support any setbacks in recovery and to develop longer term goals.

Each health professional (physiotherapist, occupational therapist or nurse) who delivered the experimental intervention attended a 1-day training session, which taught the components and format of the programme. To promote compliance with the treatment protocol, the Central Trial Team had regular contact with clinical team members, reviewing the first HIP HELPER sessions for intervention fidelity and held monthly meetings regarding study processes.

### Data collection
At the time of enrolment, sites checked eligibility and recorded demographic characteristics in the screening log. Baseline assessments were undertaken after consent was obtained, prior to randomisation. Data collected at baseline included: hospital admission, age, sex, ethnicity, height, weight, patient cognitive impairment assessed using the AMTS,[15] medical history, American Society of Anaesthesiologists grade,[18] side of hip fracture, operative procedure and hip fracture classification. Caregiver

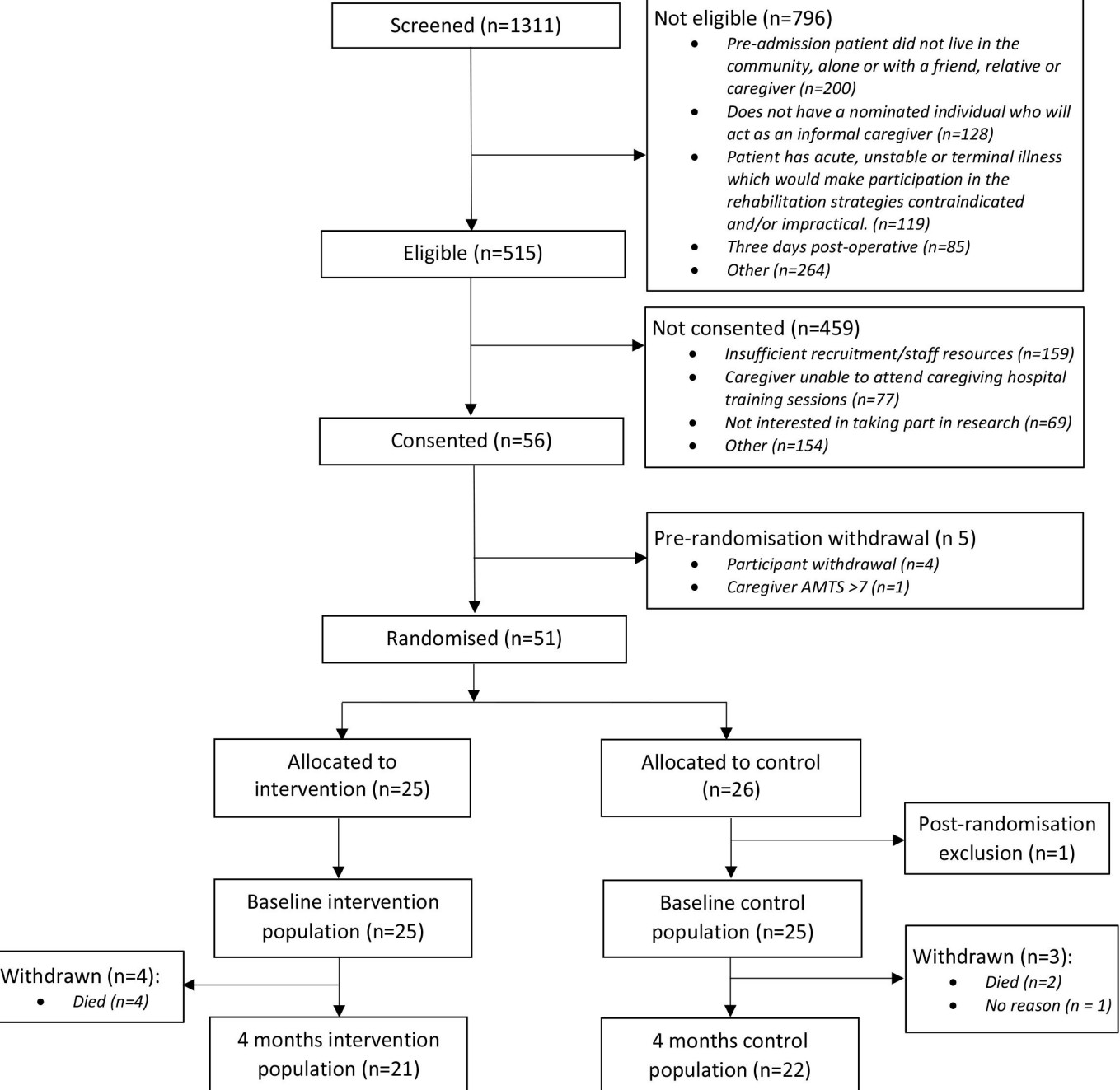

**Figure 1** Consolidated Standards of Reporting Trials diagram reporting the flow of patient–caregiver participants in the HIP HELPER study. AMTS, Abbreviated Mental Test Score.

demographic data collected included: relationship of caregiver to patient, age, sex, ethnicity, medical history, AMTS,[15] whether they lived with the patient, employment status and experience of being a caregiver (for this patient and/or for another person).

Participants were followed up at 4 months post randomisation. Data were collected via postal questionnaires by the central trial team.

## Outcome measures
To answer our feasibility objectives, we assessed:

1. Recruitment feasibility—by screening log data on: number of potential participants and their caregivers assessed for eligibility, including reasons for exclusion/non-participation, and consented to be randomised; timing and location of approach and consent.
2. Intervention acceptability—by qualitative interviews with participants and health professionals; acceptability questionnaire, study attrition at the intervention phase.
3. Intervention fidelity (healthcare professionals)—by intervention log data on: HIP HELPER session duration,

frequency, location (orthopaedic/orthogeriatric ward, rehabilitation ward or other); quality assurance (QA) to monitor HIP HELPER programme delivery.

4. Intervention fidelity (caregivers)—by caregiver HIP HELPER programme intervention logs; qualitative interviews.

5. Randomisation acceptability—by screening logs, eligibility assessment logs and consent forms; participant attrition; qualitative investigation.

6. Risk of contamination—by HIP HELPER programme log data including: QA monitoring visit checklists; delegation logs; qualitative interviews with health professionals.

7. Completeness of outcome measures—by completion rates (baseline and 4 months post randomisation). Outcome measures collected are included in online supplemental file 1.

### Randomisation and blinding

Randomisation was at the patient–caregiver dyad level (1:1 experimental and control groups) by stratification for: hospital and the presence of patient cognitive impairment (AMTS[15] < or ≥ 8 points). Sites team members performed randomisation postbaseline data collection. Allocation was concealed prior to randomisation. Randomisation was computer generated, performed by site team members on a secure, online programme, centrally administered by an independent programmer at the Norwich CTU (NCTU). The randomisation sequence was generated by NCTU programmers, tested by the trial statistician.

Due to the participatory nature of the intervention, blinding participants or the site team was not possible. Senior research team members were blinded to treatment allocation for the duration of the study.

### Sample size

We aimed to recruit 120 participants (60 patients; 60 caregivers). This was considered sufficient to answer our feasibility objectives and assess the a priori progression criteria based on Teare et al[19] recommendations.

### Data analysis and progression criteria

Consent rates, recruitment rates, attrition, missing data rates and intervention fidelity were reported as proportions with 95% CIs presented for consent and recruitment rates. The analysis of clinical outcome measures was descriptive, reported as means and SD or medians and IQR and numbers and percentages for binary and categorical variables. No formal statistical testing was undertaken.

A 'traffic light' system was used as a guide for progression to a definitive trial.[20] The progression criteria were centred around recruitment, retention, intervention fidelity and contamination.

### Study monitoring

A Trial Oversight Committee (TOC) was appointed to independently review data on safety, protocol adherence and study processes.

### Patient and public involvement

Patient involvement began during protocol development and continued throughout the study. One patient-member (not enrolled in the study) attended TOC meetings. They provided insights into the study conduct, particularly on data collection processes and helped interpret the findings to inform the study's dissemination phase.

Participants who expressed an interest in receiving information on the findings were provided with this.

### Embedded qualitative study

The aim of the embedded qualitative investigation was to assess the acceptability of the HIP HELPER programme and the research design from the perspective of caregiver dyads and health professionals. Its design was guided by Medical Research Council (MRC) guidance for evaluating complex interventions,[21–23] with the intention of understanding contextual factors influencing implementation, theorising how the HIP HELPER programme may work in practice and identifying key uncertainties to enable the programme and research design to be refined.

Six weeks after hospital discharge, caregiver dyads were invited, via a telephone call, to participate in an in-depth, semistructured interview. They were purposively sampled by age, ethnicity, prefracture disability (Nottingham ADL scale (NEADL)[24]), level of cognitive health (AMTS[15]), Disability Assessment for Dementia Scale-6[25] and study site. For health professionals, we invited for interview those who had completed the HIP HELPER programme with at least one caregiver dyad. This sample was purposively sampled by site location and clinical background, to ensure representation across physiotherapy, nursing and occupational therapy professions. All interviews were conducted virtually using Microsoft Teams or telephone. Our topic guide was informed by the MRC guidelines[22 23] and Sekhon's framework of acceptability.[26] All interviews were conducted by the same researcher (AW), an experienced postdoctoral, female, qualitative researcher. AW had no role in recruitment to the study nor intervention delivery. Interviews were audio recorded, anonymised and transcribed verbatim.

Our analysis took a two-stage approach. First deductive, to assess the quality of implementation and identify contextual factors using the MRC frameworks as a guide.[22 23] An inductive approach further explored participant's experiences and reflections on the intervention from a caregiver dyad perspective. Analysis was independently conducted by one researcher (AW) and then themed with an additional two (SHa, TS) using Reflexive Thematic Analysis, whereby the highly contextual nature of the data was acknowledged.[27 28]

**Table 1** Demographic characteristics of the patient participants at baseline

| | Intervention (n=25) | Control (n=25) |
|---|---|---|
| Gender: n (%) | | |
| Male | 4 (16.0) | 9 (36.0) |
| Female | 21 (84.0) | 16 (64.0) |
| Age in years: mean (SD) | 81.4 (8.1) | 77.6 (8.6) |
| Ethnicity: n (%) | | |
| White British | 22 (88.0) | 23 (92.0) |
| White Irish | 1 (4.0) | 1 (4.0) |
| Indian | 0 | 1 (4.0) |
| Bangladeshi | 2 (8.0) | 0 |
| Height (cm): mean (SD) | 164.2 (10.6) | 166.81 (10.0) |
| Weight (kg): mean (SD) | 65.1 (13.9) | 72.6 (17.2) |
| BMI: mean (SD) | 24.1 (4.5) | 26.1 (6.2) |
| AMTS score: median (IQR) | 10 (9–10) | 10 (9–10) |
| AMTS category | | |
| Cognitive impairment (score<8) | 5 (20.0) | 5 (20.0) |
| No cognitive impairment (score≥8) | 20 (80.0) | 20 (80.0) |
| Side of hip fracture: n (%) | | |
| Left | 10 (40.0) | 15 (60.0) |
| Right | 15 (60.0) | 10 (40.0) |
| Hip fracture classification: n (%) | | |
| Intracapsular | 14 (56.0) | 17 (68.0) |
| Intertrochanteric | 8 (32.0) | 5 (20.0) |
| Subtrochanteric | 3 (12.0) | 3 (12.0) |
| Operative procedure: n (%) | | |
| Hemiarthroplasty | 10 (41.7) | 15 (60.0) |
| THR | 2 (8.3) | 1 (4.0) |
| Cannulated screws | 3 (12.5) | 1 (4.0) |
| DHS | 4 (16.7) | 5 (20.0) |
| Intramedullary device | 5 (20.8) | 3 (12.0) |
| Missing | 1 | 0 |
| Length of hospital stay (days): median (IQR) | 15 (10–19) | 11 (8–17) |
| ASA grade: median (IQR) | 3 (3–3) | 3 (2–3) |
| Missing | 1 | 1 |
| Current medical diagnoses: n (%) | | |
| Cardiac | 8 (32.0) | 5 (20.0) |
| Asthma | 2 (8.0) | 1 (4.0) |
| COPD | 6 (24.0) | 7 (28.0) |
| Hypertension | 12 (48.0) | 14 (56.0) |
| Diabetes | 2 (8.0) | 5 (20.0) |
| Stroke | 0 | 2 (8.0) |
| Cancer | 7 (28.0) | 3 (12.0) |

Continued

**Table 1** Continued

| | Intervention (n=25) | Control (n=25) |
|---|---|---|
| Osteoarthritis | 5 (20.0) | 3 (12.0) |
| Low back pain | 4 (16.0) | 3 (12.0) |
| Depression | 0 | 0 |
| Anxiety | 0 | 0 |
| Dementia | 2 (8.0) | 2 (8.0) |
| Other | 12 (48.0) | 11 (44.0) |

AMTS, Abbreviated Mental Test Score; ASA, American Society of Anesthesiologists; BMI, body mass index; cm, centimetres; DHS, dynamic hip screw; kg, kilograms; OPD, chronic obstructive pulmonary disease; THR, total hip replacement.

## RESULTS

### Patient characteristics and treatment

As a result of disruption on NHS services caused by the COVID-19 pandemic, we recruited 102 participants (51 patients; 51 caregivers) from April 2021 to February 2022.

A summary of the patient-cohort characteristics is presented in table 1. Seventy-four per cent (37/50) were female, with a mean age of 81.4 years in the intervention group and 77.6 years in the control group. In total, 94% (47/50) were white British or Irish. Ten patient participants (five per group) had an AMTS[15] of less than 8, indicating cognitive impairment at baseline. The median length of hospital stay was 15 days (IQR: 10–19) in the intervention group and 11 days in the control group (IQR: 8–17). As table 1 demonstrates, people with hip fracture in the intervention group were older, with more medical comorbidities and more frequently presented with intratrochanteric fractures.

A summary of the caregiver–participant characteristics is presented in table 2. Fifty-three per cent (36/50) of the cohort were female. Mean age of caregivers in the intervention group was 66 years and 58 years in the control group. Caregivers were most frequently patient-participant's children (53%; 26/50) or a spouse (37%; 18/50). Most caregivers were not working (65%; 32/50); 20% (10/50) were in full-time work.

### Feasibility outcomes

The outcomes of the progression criteria traffic-light assessment are presented in table 3.

### Recruitment, retention and randomisation acceptability

The CONSORT flow chart is presented in figure 1. As this illustrates, 1311 potential participants were screened. Of these, 515 (39%; 95% CI 37% to 42%) were eligible, with 56/515 (11%; 95% CI 8% to 13%) of eligible participant dyads consented to participate. Five participant dyads were withdrawn prior to randomisation. A summary of reasons for being ineligible or being eligible but not consenting is presented in online supplemental files 2

**Table 2** Demographic characteristics of the caregiver participants at baseline

| | Intervention (n=25) | Control (n=25) |
|---|---|---|
| **Gender: n (%)** | | |
| Male | 14 (56.0) | 9 (37.5) |
| Female | 11 (44.0) | 15 (62.5) |
| Missing | 0 | 1 |
| **Age in years: mean (SD)** | 66.2 (13.6) | 57.7 (12.9) |
| Missing | 1 | 2 |
| **Ethnicity: n (%)** | | |
| White British | 20 (80.0) | 23 (95.8) |
| White Irish | 1 (4.0) | 0 |
| White—other | 1 4.0) | 1 (4.2) |
| Mixed—other | 1 (4.0) | 0 |
| Bangladeshi | 2 (8.0) | 0 |
| Missing | 0 | 1 |
| **AMTS score: median (IQR)** | 10 (10–10) | 10 (10–10) |
| Missing | 0 | 1 |
| **Relationship to participant: n (%)** | | |
| Spouse | 10 (40.0) | 8 (33.3) |
| Daughter/son | 13 (52.0) | 13 (54.2) |
| Grandchild | 0 | 1 (4.2) |
| Other | 2 (8.0) | 2 (8.3) |
| Missing | 0 | 1 |
| **Caregiver living with participant: n (%)** | | |
| Yes | 16 (69.6%) | 14 (60.9%) |
| No | 7 (30.4%) | 9 (39.1%) |
| Missing | 2 | 2 |
| **Occupation: n (%)** | | |
| Not working | 17 (68.0) | 15 (62.5) |
| Part-time | 3 (12.0) | 4 (16.7) |
| Full-time | 5 (20.0) | 5 (20.8) |
| Missing | 0 | 1 |
| **Current medical diagnoses: n (%)** | | |
| Cardiac | 2 (8.0) | 1 (4.0) |
| Asthma | 0 | 4 (16.0) |
| COPD | 0 | 0 |
| Hypertension | 2 (8.0) | 1 (4.0) |
| Diabetes | 1 (4.0) | 2 (8.0) |
| Stroke | 0 | 0 |
| Cancer | 1 (4.0) | 0 |
| Osteoarthritis | 3 (12.0) | 1 (4.0) |
| Low back pain | 3 (12.0) | 4 (16.0) |
| Depression | 0 | 3 (12.0) |
| Anxiety | 0 | 6 (24.0) |
| Other | 3 (12.0) | 2 (8.0) |

AMTS, Abbreviated Mental Test Score; COPD, chronic obstructive pulmonary disease.

and 3. Recruitment activity per site is presented in online supplemental file 4.

At 4-month follow-up, 43/51 participant dyads (86%) (21 intervention, 22 control) remained in the study. Six participants died, one withdrew without reason; in each instance, the complete dyad was withdrawn. At 4 months, there were 8 patient participants with cognitive impairment, 35 without cognitive impairment. The groups were largely comparable at baseline (table 1).

### Intervention fidelity (health professionals)

Online supplemental file 5) illustrate the delivery of the hospital-based and telephone-based HIP HELPER sessions. As summarised in online supplemental 6, 12/25 participant dyads (48%) of participant dyads received the minimal compliance level of all three HIP HELPER inpatient sessions and one telephone call. Reasons for non-compliance were: insufficient staff to deliver the intervention due to staff redeployment and interruption of service provision or visiting of participants due to the COVID-19 pandemic (n=2); patient participant transferred from a unit (n=2); death (n=4); treatment discontinuation (n=4); or did not answer the telephone (n=3).

Table 4 illustrates that all components of the HIP HELPER programme were delivered during testing. Components which were most frequently delivered were: explanation on recovery expectations (96%; 23/25), goal setting (92%; 22/25) and pacing and behaviour modification (92%; 22/25). Less frequently delivered components were the more functionally demanding activities such as washing, dressing and stair and car transfers (38%; 9/25 each).

### Intervention fidelity (caregivers)

Only two caregiver–participants returned their caregiver log. Accordingly, there were insufficient data to permit robust assessment of intervention fidelity from the caregiver perspective. This was therefore not analysed.

### Contamination

From the qualitative investigations, case report forms for treatment received, protocol deviation reports and delegation logs of treating health professionals, there was no evidence of between-group intervention contamination.

### Outcome data response rate

There was limited difference in the completion of the caregiver–participant outcomes at baseline or 4 months in either group (online supplemental files 7 and 8). However, there was a notable difference in outcome completion at 4 months for patients with cognitive impairment and their caregivers. While 80% of caregivers in the control group completed the majority of outcomes, only two caregivers of people with cognitive impairment completed the outcomes (EQ-5D proxy only) in the intervention group. Patient participants in the intervention group reported a higher response rate to all outcomes at 4 months except the NEADL[24] (online supplemental file 7).

**Table 3** Progression criteria traffic-light summary table

| | Green (Go) | Amber (Amend) | Red (Stop) | Judgement |
|---|---|---|---|---|
| Recruitment | > 40% of patients screened would be eligible | 30%–40% would be eligible | < 30% would be eligible | 39% participants were eligible |
| Randomisation acceptability | > 40% of eligible patients consent to be randomised | 20%–40% would be randomised | < 20% would be randomised | 11% of eligible participants were randomised |
| Intervention fidelity (healthcare professionals) | > 70% of participants compliant with their allocated intervention as randomised | 50%–70% received intervention as randomised | < 50% received intervention as randomised | 48% received 'complete intervention' as randomised, limited by COVID-19 |
| Intervention fidelity (caregivers) | > 90% of participants adopted HIP HELPER intervention post-discharge | 60%–90% adopted HIP HELPER post-discharge | < 60% adopted HIP HELPER post-discharge | Unable to assess with insufficient caregiver logs |
| Contamination | < 5% of participants in either group received majority of their allocated treatment crossover | 5%–10% of participants crossover | > 10% of participants crossover | 0% evidence of contamination |

**Table 4** Table illustrating the frequency to-which the components of the HIP HELPER intervention were delivered to participants

| | Intervention (N=25) | | | |
|---|---|---|---|---|
| Item | Session 1 N (%) | Session 2 N (%) | Session 3 N (%) | At least one occurrence during sessions 1–3 |
| **Practical skills transfers:** | | | | |
| Bed to chair | 16 (66.7) | 13 (68.4) | 8 (57.1) | 20 (83.3) |
| Toilet | 4 (16.7) | 7 (36.8) | 4 (28.6) | 12 (50.0) |
| Walking and walking aids | 15 (62.5) | 12 (63.2) | 9 (64.3) | 19 (79.2) |
| In/out bed | 16 (66.7) | 12 (63.2) | 8 (57.1) | 20 (83.3) |
| Car | 3 (12.5) | 6 (31.6) | 2 (14.3) | 9 (37.5) |
| Stairs | 3 (12.5) | 6 (31.6) | 3 (21.4) | 9 (37.5) |
| Goal setting theory | 21 (87.5) | 13 (68.4) | 4 (28.6) | 22 (91.7) |
| Goal setting practice | 15 (62.5) | 15 (79.0) | 6 (42.9) | 22 (91.7) |
| Pacing and behaviour theory | 20 (83.3) | 9 (47.4) | 6 (42.9) | 22 (91.7) |
| Pacing and behaviour task | 13 (54.2) | 10 (52.6) | 5 (35.7) | 18 (75.0) |
| Expectations of recovery pathways | 22 (91.7) | 9 (47.4) | 5 (35.7) | 23 (95.8) |
| **Practical skills:** | | | | |
| Washing | 1 (4.2) | 7 (36.8) | 1 (7.1) | 9 (37.5) |
| Dressing | 1 (4.2) | 9 (47.4) | 1 (7.1) | 9 (37.5) |
| **Caregiver:** | | | | |
| Management discussion | 3 (12.5) | 7 (36.8) | 14 (100) | 18 (75.0) |
| Pacing discussion | 4 (16.7) | 6 (31.6) | 10 (71.4) | 16 (66.7) |
| Case scenario discussion | 1 (4.2) | 3 (15.8) | 9 (64.3) | 12 (50.0) |
| Provision and discussion on HIP HELPER manual | 15 (62.5) | 4 (21.1) | 10 (71.4) | 22 (91.7) |
| Confirmation of HIP HELPER telephone calls | 0 | 0 | 13 (92.9) | 13 (54.2) |
| Other 1: ('pain relief', 'lack of hip precautions', 'caregiver questions', 'equipment ordering') | 1 (4.2) | 2 (10.5) | 1 (7.1) | 2 (8.3) |
| Other 2: ('equipment ordering') | 0 | 0 | 1 (7.1) | 1 (4.2) |
| Other 3: ('sleep') | 0 | 0 | 1 (7.1) | 1 (4.2) |

Number of planned sessions not performed were: 1 Session 1, 6 Session 2, 11 Session 3; missing data not given, percentages are out of non-missing data.

## Clinical outcomes

Online supplemental file 9 illustrates the descriptive clinical outcomes presented as median and IQRs for baseline and 4-month follow-up. Between-group differences should be interpreted with caution given the level of missing data in both groups (online supplemental file 7), a potential baseline difference between the groups for age, presented comorbidities and fracture type (table 1) and underpowered analyses.

No participant, from either group, experienced a related adverse event or serious adverse event. A summary of the patient–caregiver reported adverse events is presented in online supplemental file 10.

Intervention acceptability questionnaire data indicated the HIP HELPER programme was regarded as acceptable by people with hip fracture (online supplemental file 11) and caregivers (online supplemental file 12).

## Qualitative study

Fourteen caregiver dyads were invited to be interviewed. Ten agreed to participate (intervention: seven participants; control: three participants). All eight health professionals approached, agreed to be interviewed. Online supplemental file 13 summarises the patient-caregiver's and health professional's characteristics.

Our findings are grouped into three main themes: context, intervention delivery and study procedures.

## Context

This study was conducted during the COVID-19 pandemic. Patient visitor policies and restrictions were in place. Most caregiver dyads suggested that the opportunity to visit their friend/relative was a main driver for participation.

> This trial helped me visit **** once or twice more during her stay in hospital than I would have been allowed to. (Caregiver 1, Intervention Group, Male, Site 4)

For health professionals, allocating visiting time slots created a further challenge of obtaining consent.

> With visiting times, we're making sure it's the right carer coming in because it might not be them that have the slots booked that week. (Occupational Therapist, Female, Site 1).

Changes in visitor policies/restrictions would need to be considered for any future trial.

From the perspective of health professionals, one of the most common reasons for non-participation was attributed to perceived burden on caregivers.

> They say, 'oh no that seems like a lot for my daughter to take on or that seems a lot for husband to do they already do enough, or I don't think they'd manage that'. So, it's that perception that they don't want to put any more burden on someone. Seems to be the main reason we find. (Occupational Therapist, Site 1)

> Caregivers who work full-time or spouses who are too frail to make it into hospital just can't. (Research Physiotherapist, Male, Site 5)

For this study, there was the added complexity of dyad recruitment, as reflected in this comment:

> I think getting both the caregiver and the patient consent is a bit of a headache. (Occupational Therapist, Female, Site 1)

Initially, when health professionals approached the person with hip fracture, a key reason for decline was concerns about feelings of burden on their caregivers. Staff felt recruitment was more successful when the potential caregiver was approached first.

## Intervention delivery

Participants perceived the workbook to be helpful in giving a sense of tangible timeframes for recovery. Goal setting was seen as helping in pushing people out of their comfort zones and allowed them to reflect on progress.

> Made us gauge our progress and that he wanted our response to what goals we had. And as I said, going back through it each time we read another page, you realise that we've upped the goal and how far we've progressed (Caregiver, Intervention Group, Male, Site 4)

Areas of workbook refinement were identified, such as the volume of information included. Importantly, some felt that the workbook did not reflect the life circumstances of younger participants and those still in work.

> Like, say, when I did it [the case studies], I was like how am I gonna drag this out for an hour with a patient who can just about get bed to chair. (Physiotherapist, Male, Site 5)

The follow-up phone calls were seen as helpful. For health professionals, this addition added a rewarding element to their role.

> Telephone calls have been really useful. Especially for me because I work on inpatients where don't often get time to follow-up a patient, see how they are, see if there's any concerns. I suppose, from a development point of view, knowing what has worked and what hasn't. (Physiotherapist, Male, Site 2)

For the patient–caregiver dyads, telephone calls provided encouragement and reassurance to maintain and progress activities.

> Reassured me that I'm doing the right things and where I should expect to be. (Person with Hip Fracture, Intervention Group, Female, Site 4)

Participants perceived value in the telephone calls particularly in navigating additional services and support after discharge. They expressed this would have been challenging without this follow-up.

## Study procedures

Respondents repeatedly acknowledged the perceived burden of completing the outcome measures.

> It would be a lot for say, the husband to stay at home, trying to fill in 15 pages, when the wife is not there and then they have to try to adjust living by themselves and with all this happening. (Nurse Practitioner, Female, Site 2)

The outcome measures evaluated patient–caregiver outcomes from a biopsychosocial perspective.[29] Accordingly, some patient reported outcomes posed questions regarding an individual's ability to cope with the physical and psychosocial challenges of trauma. For some respondents, this was reported as emotionally difficult.

> So, when you were doing it [questionnaires] with the carer, there was a couple of times where it was uncomfortable. They might not want to say that I have low in mood in front of someone. It was quite upsetting for the carer to divulge that information with you, or to bring up things from their past as well. (Physiotherapist, Male, Site 3)

## DISCUSSION

The findings from this feasibility study indicate that the HIP HELPER intervention was acceptable to patients, informal caregivers and health professionals but the trial design requires further development to ensure feasibility. Modifications should be made on promoting intervention adherence, prioritising outcome measures to test future effectiveness of data completeness and exploring strategies to support the recruitment of patients with cognitive impairment.

The completion of data from the outcomes at 4 months was lower than anticipated. This was particularly for participants with cognitive impairment. The qualitative study indicated that participants found the number of outcome measures challenging to complete and future study could better discern what outcome measures are important to people following hip fracture and their caregivers.[30] Streamlining should be made to determine the outcomes which are most valuable to participants and clinical commissioners. A second major modification relates to the acceptability of randomisation. Only 11% of eligible participants were randomised. The qualitative study indicated this may have been because patient participants did not wish to 'burden their caregiver' with the study when they were initially approached and so declined participation. We originally designed the study approach pre-COVID 19 with an initial approach occurring when both patient and caregiver were together, during visiting hours. This was to facilitate a collaborative decision between the dyad, rather than one member deciding participation. However, due to COVID-19 restrictions on visiting, this was not possible. The qualitative study indicated that the originally planned approach should have been more successful. We recommend that both members of the dyad should be approached simultaneously in future trials, to mitigate such low conversion to randomisation.

A written, information guide about rehabilitation, recovery goals and caregiver responsibilities in the home has been previously reported as valuable to other populations.[31] The findings of this study indicate that while the addition of this intervention was beneficial, the HIP HELPER workbook received mixed views from participants. The level of detail, degree of context and order of material covered in the HIP HELPER workbook was considered by many participants as too great. Equally, the qualitative findings suggested that the current materials were not representative of all patients and caregivers, most notably younger people who sustain a hip fracture. Further patient and public consultation with the research and clinical hip fracture community is needed to modify this workbook and associated digital offerings of this material.

Approximately 40% of people who sustain a hip fracture present with dementia.[32] While previous authors have acknowledged potential challenges in recruiting people with dementia to drug trials,[33] no studies have explored recruitment expectations or strategies to address low recruitment to non-pharmacological interventions.[34] We anticipated recruiting 20 patient participants with cognitive impairment. In total, 10 participants were recruited with mild cognitive impairment. The qualitative findings suggest that offering further support to research site members who approach patients with cognitive impairment and their caregivers, to promote skills conveying study information, may be beneficial. Furthermore, given the poor response rate in 4-month outcome data for these participants, consideration on the appropriateness of the current instruments used and model of delivery of outcome battery for people with cognitive impairment and their caregiver, should be considered in future trials of this population.[35]

Previous evidence suggests that health professionals have been inflexible about people with hip fracture and their caregivers to discuss care plans, when these do happen.[36–38] The qualitative study highlighted that those participants who received the HIP HELPER programme appreciated the contact and opportunity to explore skills and knowledge for early recover and caregiver support following hip fracture. The addition of the telephone calls was reported as offering beneficial, additional, post-discharge support in a flexible approach. However, intervention fidelity was lower than anticipated. Unfortunately, it is difficult to separate the challenges which COVID-19 placed on research conduct and service provision and challenges in delivering the HIP HELPER programme in a non-health crisis. Sites were challenged in delivering the intervention due to staffing, patient transfers, visiting restrictions and earlier than planned discharges. This impacted on fidelity of the 'full' HIP HELPER programme to all participants. Deeper exploration on modification

to intervention delivery and what components are 'core' ingredients to the programme to estimate compliance thresholds would be warranted.

This study presented with strengths and limitations. A notable strength was the ability to recruit over 100 participants from five NHS organisations during the 2020 COVID-19 pandemic. While a short-fall of 10 participant dyads, given the challenges in managing site opening and research conduct during the COVID-19 pandemic, the ability to undertake this was considered a success. In the absence of COVID-19 (or such like) restrictions on visiting, patient flow and research activity in NHS settings, we would anticipate that this impact would be negated in future trials of this population. Second, although we planned to assess whether caregivers adopted their caregiving knowledge from the intervention into the home environment, only two participants returned these data. This was a major limitation and resulted in an inability to answer an a prior progression criterion. The findings from the qualitative study and acceptability questionnaires may suggest carryover of the intervention into practice. However, we acknowledge that this does not offer the granularity of detail which the original caregiver log would have conveyed. Finally, the follow-up rates and data competition for clinical outcomes were low. Accordingly, it was not possible to confidently assess for a signal of efficacy in the experimental intervention. Further modifications in what and how clinical outcome data are collected should be considered as part of the following work to improve the feasibility of this trial design.

## CONCLUSIONS

The HIP HELPER programme was acceptable to participants and health professions. Further modifications to the trial design are needed to ensure feasibility. These findings will form the basis of reflection and refinement to the trial design to test the clinical and cost-effectiveness of the programme in addition to understand the scalability and pathway to implementation.

**Author affiliations**
[1]Warwick Medical School, University of Warwick, Coventry, UK
[2]School of Health Sciences, University of East Anglia, Norwich, UK
[3]Norwich Medical School, University of East Anglia, Norwich, UK
[4]School of Education, University of East Anglia, Norwich, UK
[5]Nuffield Department of Orthopaedics, Rheumatology and Musculoskeletal Sciences, University of Oxford, Oxford, UK
[6]Department of Clinical Gerontology and Geriatric Rehabilitation, Robert Bosch Hospital, Stuttgart, Germany
[7]School of Medicine, University of Nottingham, Nottingham, UK
[8]College of Medicine and Public Health, Flinders University, Adelaide, South Australia, Australia
[9]College of Medicine and Health, University of Exeter, Exeter, UK

**Collaborators** The HIP HELPER Study Collaborators: Penny Clifford (Norfolk, PPI Representative), Lis Freeman (Norfolk, PPI Representative), Rene Gray (Principal Investigator—James Paget University Hospital NHS Trust), Yan Cunningham (Principal Investigator—City Hospitals Sunderland NHS Foundation Trust), Sarah Langford (Principal Investigator—Northumbria Healthcare NHS Foundation Trust), Dr Mark Baxter (Principal Investigator—University Hospital Southampton NHS Foundation Trust), Jessica Pawson—(Principal Investigator—Barts Health NHS Trust), Melissa Taylor (James Paget University Hospital NHS Trust), Anna Mellows (James Paget University Hospital NHS Trust), Kate Lacey (James Paget University Hospital NHS Trust), Alex Herring (City Hospital Sunderland NHS Foundation Trust), Diane Williams (Northumbria Healthcare NHS Foundation Trust), Anna Cromie (Northumbria Healthcare NHS Foundation Trust), Gail Menton (Northumbria Healthcare NHS Foundation Trust), Warren Corbett (University Hospital Southampton NHS Foundation Trust), Helen Jowett (University Hospital Southampton NHS Foundation Trust), Vishwanath Joshi (Barts Health NHS Trust), Maninderpal Matharu (Barts Health NHS Trust), Maria Baggot (University Hospital Southampton NHS Foundation Trust),and David Barker (University Hospital Southampton NHS Foundation Trust). Oversight Committee Membership: TOC Members: Associate Professor Susan Dutton (Chair; University of Oxford, UK), Professor Opinder Sahota (University of Nottingham, UK), Dr Katie Sheehan (Kings College London, UK).

**Contributors** TOS, SHa, SHo, ABC, KG, P-AA, KP, PL, MLC, SL and MC researched the topic and devised the study. TOS, SHa, P-AA, RK, ABC, KG, SHo, AW, KP, PL, MLC, SL, MC provided the first draft of the manuscript. ABC provided statistical oversight. TOS, SHa, P-AA, RK, ABC, KG, AW, SHo, KP, PL, MC, SL and MLC contributed equally to manuscript preparation. TOS acts a guarantor.

**Funding** This project is funded by the National Institute for Health and Care Research (NIHR) under its Research for Patient Benefit (RfPB) Programme (Grant Reference Number NIHR200731). SL role in this study was supported by the National Institute for Health and Care Research Exeter Biomedical Research Centre. The views expressed are those of the author(s) and not necessarily those of the NIHR or the Department of Health and Social Care.

**Disclaimer** MLC and SHo are supported by the National Institute for Health Research (NIHR) Oxford Biomedical Research Centre (BRC). SL is supported by the National Institute for Health and Care Research Exeter Biomedical Research Centre. The views expressed are those of the author(s) and not necessarily those of the NHS, the NIHR or the Department of Health and Social Care.

**Competing interests** None declared.

**Patient and public involvement** Patients and/or the public were involved in the design, or conduct, or reporting, or dissemination plans of this research. Refer to the Methods section for further details.

**Patient consent for publication** Not applicable.

**Ethics approval** This study involves human participants and was approved by NHS NRES ethical committee—North East—Newcastle & North Tyneside 1 Research Ethics Committee (20/NE/0213) Date: 16 March 2021. Participants gave informed consent to participate in the study before taking part.

**Provenance and peer review** Not commissioned; externally peer reviewed.

**Data availability statement** Data are available upon reasonable request. Data includes access to the full protocol, anonymised participant level dataset and statistical code. Access to the de-identified dataset for purposes of research other than this study, would be at the discretion of the Chief Investigator, TOS and Norwich CTU. Requests for the de-identified dataset generated during the current study should be made to the Chief Investigator, TOS (email: toby.o.smith@warwick.ac.uk) or Norwich CTU (NorwichCTU@uea.ac.uk). TOS and Norwich CTU will consider requests once the main results from the study have been published up until 31st December 2028.

**ORCID iDs**
Toby O Smith http://orcid.org/0000-0003-1673-2954
Sarah Hanson http://orcid.org/0000-0003-4751-8248
Allie Welsh http://orcid.org/0000-0001-8278-6673
Sally Hopewell http://orcid.org/0000-0002-6881-6984

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
