## [Reviewer comments · BMJ Open]

ARTICLE DETAILS

TITLE (PROVISIONAL)	Hospital-based caregiver intervention for people following hip fracture surgery (HIP HELPER): multi-centre randomised controlled feasibility trial with embedded qualitative study in England.
AUTHORS	Smith, Toby; Khoury, Reema; Hanson, Sarah; Welsh, Allie; Grant, Kelly; Clark, Allan; Ashford, Polly-Anna; Hopewell, Sally; Pfeiffer, K; Logan, Phillipa; Crotty, Maria; Costa, Matthew; Lamb, Sarah

VERSION 1 – REVIEW

REVIEWER	Adili, Anthony McMaster University Faculty of Health Sciences, Surgery
REVIEW RETURNED	25-Jul-2023

GENERAL COMMENTS	Thank you for the opportunity to review this interesting pilot trial on a hospital-based caregiver intervention for people following hip fracture surgery. The study is well-designed and well-written. I only have minor comments below. 1. Were there any changes from the published protocol?2. It is unclear who did the training and follow-up. Did the health professional have to do this training on top of their usual duties? I would think that's difficult to schedule in, particularly if this is to be scaled up. Am I interpreting this correctly that the intervention lasted 3 hours (3x60min sessions)? That doesn't seem scalable and would require a ton of extra staff.3. Is there justification for this sample size? Thabane et al suggested quantitative way of determining pilot trial sample size based on feasibility objectives. https://bmcmedresmethodol.biomedcentral.com/articles/10.1186/1471-2288-10-14. In the Recruitment, retention and randomisation acceptability section, the authors shouldn't conclude one group is older. The numbers are small and there is no significance testing.5. In the Outcome Data Response Rate section, the intervention patients had a higher response rate to all questionnaires, but the next sentence says the control group had higher response rate for NEADL6. Table 1 says "intra-trochanteric fracture". Do you mean inter-trochanteric?7. There are a lot of supplementary files. Some could be combined, condensed, or removed. E.g. suppl3 is redundant with the flow diagram, 4,6,7,8 could be summarized in one table.
---

VERSION 1 – AUTHOR RESPONSE

Reviewer: 1

Comment: Thank you for the opportunity to review this interesting pilot trial on a hospital-based caregiver intervention for people following hip fracture surgery. The study is well-designed and well-written. I only have minor comments below.

Response: Thank you for the comments raised. We have addressed the points raised and itemised these changes below.

Comment: 1. Were there any changes from the published protocol?

Response: The only change was the introduction of an online approach to deliver the HIP HELPER programme in response to the COVID-19 pandemic. This was only enacted for one participant. This has now been included in the revised manuscript (Methods, Paragraph 1, Line 2-5).

Comment: 2. It is unclear who did the training and follow-up. Did the health professional have to do this training on top of their usual duties? I would think that's difficult to schedule in, particularly if this is to be scaled up. Am I interpreting this correctly that the intervention lasted 3 hours (3x60min sessions)? That doesn't seem scalable and would require a ton of extra staff.

Response: We have added information on which health professionals did the training, delivered the intervention (Methods, Study Treatments, Paragraph 4, Line 1) and who lead the follow-up data collection (Methods, Data Collection, Paragraph 2, Lines 1-2). The question on scalability is important. An objective of this feasibility study was to test the acceptability and deliverability of the intervention. We have included information on scalability for future study as suggested (Conclusion, Line 3-4).

Comment: 3. Is there justification for this sample size? Thabane et al suggested quantitative way of determining pilot trial sample size based on feasibility objectives.
<https://bmcmedresmethodol.biomedcentral.com/articles/10.1186/1471-2288-10-1>

Response: We have now provided the citation which formed the basis of the sample size estimate when designing the study (Methods, Sample Size, Paragraph 1, Line 2-3).

Comment: 4. In the Recruitment, retention and randomisation acceptability section, the authors shouldn't conclude one group is older. The numbers are small and there is no significance testing.

Response: We have amended this as suggested (Results, Recruitment, Retention and Randomisation, Paragraph 2, Lines 3-5).

Comment: 5. In the Outcome Data Response Rate section, the intervention patients had a higher response rate to all questionnaires, but the next sentence says the control group had higher response rate for NEADL

Response: Apologies. We have clarified this as all outcomes except the NEADL had a higher response rate in the intervention group (Supplementary File 7)(Results, Outcome Data Response Rate, Paragraph 1, Line 6-8).

Comment: 6. Table 1 says "intra-trochanteric fracture". Do you mean inter-trochanteric?

Response: This has been corrected (Table 1).

Comment: 7. There are a lot of supplementary files. Some could be combined, condensed, or removed. E.g. suppl3 is redundant with the flow diagram, 4,6,7,8 could be summarized in one table.

Response: Thank you. We have reviewed this and feel that the information conveyed in each table is different and attempting to merge these into condensed tables could be challenging for readers to understand. Accordingly, we have kept the format of these tables as they are currently. Given that these are Supplementary Files, we feel this would not interrupt the flow of the paper however, if the reviewer and editorial team feel strongly about this, we would be happy to re-review this current position.

VERSION 2 – REVIEW

REVIEWER	Adili, Anthony McMaster University Faculty of Health Sciences, Surgery
REVIEW RETURNED	10-Nov-2023
GENERAL COMMENTS	Thank you, the authors have sufficiently addressed my comments.

VERSION 2 – AUTHOR RESPONSE